# Effect of A-Site Nonstoichiometry on Defect Chemistry and Electrical Conductivity of Undoped and Y-Doped SrZrO_3_

**DOI:** 10.3390/ma12081258

**Published:** 2019-04-17

**Authors:** Liliya Dunyushkina, Adelya Khaliullina, Anastasia Meshcherskikh, Alexander Pankratov, Denis Osinkin

**Affiliations:** Institute of High Temperature Electrochemistry, 20 Akademicheskaya St, 620990 Ekaterinburg, Russia; adelia01@mail.ru (A.K.); lazyty@mail.ru (A.M.); a.pankratov@ihte.uran.ru (A.P.); osinkinda@mail.ru (D.O.)

**Keywords:** Y-doped strontium zirconate, yttrium solubility, strontium nonstoichiometry, electrical conductivity, proton conduction, DRT

## Abstract

The effect of Sr-nonstoichiometry on phase composition, microstructure, defect chemistry and electrical conductivity of Sr_x_ZrO_3−δ_ and Sr_x_Zr_0.95_Y_0.05_O_3−δ_ ceramics (SZx and SZYx, respectively; x = 0.94–1.02) was investigated via X-ray diffraction, scanning electron microscopy, energy-dispersive X-ray spectroscopy and impedance spectroscopy followed by distribution of relaxation times analysis of impedance data. It was shown that at low Sr deficiency (x > 0.96 and 0.98 for SZx and SZYx, respectively) a solid solution of strontium vacancies in strontium zirconate crystal structure forms, whereas at higher Sr deficiency the secondary phase, zirconium oxide or yttrium zirconium oxide, is precipitated. Yttrium solubility limit in strontium zirconate was found to be close to 2 mol%. Y-doped strontium zirconates possess up to two orders of magnitude higher total conductivity than SZx samples. A-site nonstoichiometry was shown to have a significant effect on the electrical conductivity of SZx and SZYx. The highest total and bulk conductivity were observed at x = 0.98 for both systems. Increasing the conductivity with a rise in humidity indicates that proton conduction appears in the oxides in wet conditions. A defect model based on consideration of different types of point defects, such as strontium vacancies, substitutional defects and oxygen vacancies, and assumption of Y ions partitioning over Zr and Sr sites was elaborated. The proposed model consistently describes the obtained data on conductivity.

## 1. Introduction

Perovskite oxides exhibiting high proton conductivity are promising electrolyte materials for solid oxide fuel cells (SOFCs) which convert the chemical energy of fuel directly into electrical energy with high efficiency and low air pollutants emissions [1,2]. Using proton conducting electrolytes allows reducing SOFC operation temperature to the intermediate temperature range (500–700 °C) due to the lower activation energy of proton mobility comparing to conventional oxygen-ion electrolytes [3]. Among the perovskite oxides, acceptor-doped barium and strontium cerates exhibit the proton conductivity sufficiently high for practical applications at intermediate temperature range but their chemical stability in CO_2_ and water containing atmospheres is poor [4,5,6,7]. In contrast, zirconate-based perovskites (AZrO_3_, A = Ba, Sr) combines high chemical stability with satisfactory proton conductivity [7,8].

Proton conductivity of strontium zirconate can be enhanced by substitution of Zr by acceptor dopants, such as Yb, Y and Dy, due to the formation of large concentrations of oxygen vacancies as charge-compensating centers [9,10,11]. Y-doped SrZrO_3_ was reported to be a good protonic conductor exhibiting its maximum proton conductivity at the dopant concentration of 5 mol% [11,12,13].

A-site nonstoichiometry in zirconate-based oxides was shown to affect their transport properties, however, the reported results are rather contradictory. A-site deficiency of 2 mol% in SrZr_0.9_Dy_0.1_O_3−δ_ slightly improved the total conductivity attributed to enhanced sinterability of ceramic samples [10]. A-site deficiency in undoped Ca_1−x_ZrO_3_ (x > 0.01) was found to result in a significant decrease of electrical conductivity mostly due to increasing blocking behavior of grain boundaries [14]. Barium deficiency in yttrium-doped barium zirconate was shown to have a detrimental impact on the proton conductivity presumably because of dopant partitioning over both A and B sites [15]. Therefore, the role of A-site nonstoichiometry in establishing the defect chemistry and proton conductivity of alkaline earth zirconates is still not clear.

The purpose of this research was to clarify the effect of Sr-nonstoichiometry on phase composition, microstructure, defect chemistry and electrical conductivity of Y-doped SrZrO_3_. Oxides with the nominal compositions of Sr_x_Zr_0.95_Y_0.05_O_3−δ_ (SZYx) with x = 0.94–1.02 were chosen for investigation. In addition, undoped strontium zirconates, Sr_x_ZrO_3−δ_ (SZx, x = 0.94–1.00), were also studied to distinguish the role of yttrium substitution on charge transport. The phase and chemical composition of the samples were characterized by X-ray diffraction (XRD), scanning electron microscopy (SEM) and energy-dispersive X-ray spectroscopy (EDX). The electrical conductivity was measured by using impedance spectroscopy. Distribution of relaxation times (DRT) analysis of the impedance data was used to identify the bulk and grain boundary contributions to the total conductivity.

## 2. Materials and Methods

SZx and SZYx powders were synthesized by chemical solution method using ZrOCl_2_·8H_2_O, Y(NO_3_)_3_·nH_2_O and SrCO_3_ (all with 99% purity) as precursors. Solutions of zirconium oxychloride octahydrate in distilled water and yttrium nitrate hydrate in ethanol, with concentrations of 49.7 g of ZrO_2_ and 81.3 g of Y_2_O_3_ per 1 L, respectively, were prepared. The calculated amounts of these solutions and SrCO_3_ powder were mixed in the ratios corresponding to the nominal compositions. Then, citric acid was added (with a molar ratio of metal cations to citric acid 1:2). After the citric acid was dissolved, ethylene glycol was added (with a molar ratio of metal cations to ethylene glycol of 1:5). The solution was stirred and heated on a hot plate to evaporate the solvent, followed by heating to 1100 °C with 2-h dwell time. The obtained powders were ground, calcined at 1200 °C for 2 h, then reground, uniaxially compacted in hardened steel dies into 14 mm diameter pellets and then isostatically cold pressed at 130 MPa. Samples were then sintered at 1650 °C for 5 h in air.

Phase composition of the samples was examined using X-ray diffraction (XRD) with Cu Kα radiation (D-Max 2200, Rigaku, Tokyo, Japan). The XRD study was performed on the powders obtained by thorough grinding of the sintered pellets in an agate mortar. The unit cell parameters were obtained using MDI Jade 6.5 software. Accuracy of the lattice parameters determination was 0.001 Å. For microstructural and compositional investigations, the sintered pellets were polished finishing with 1 μm grit diamond paste and thermally etched at the temperatures of 1400 and 1650 °C for 4 h to make the grain boundaries visible. Several samples (SZY0.98, SZY1.00 and SZY1.02) were polished using MetPrep 4™ grinding and polishing machine (Allied High Tech Products, Inc., Compton, CA, USA) and studied without thermal etching in order to obtain information about the interior of the samples. The samples were studied using MIRA 3 LMU (Tescan, Brno, Czechia) scanning electron microscope (SEM) equipped with an energy-dispersive X-Ray spectroscopy (EDX) system (Oxford Instruments X-MAX 80, Abingdon, UK). For the chemical composition study, EDX data were collected and averaged for 10 or more spots.

The density of the sintered samples was evaluated by measuring their weight and geometrical dimensions. Theoretical density was estimated using the unit cell parameters determined by XRD and assuming the nominal compositions of the specimens. Relative densities were calculated by dividing the measured density by the theoretical density. The uncertainty on relative density was estimated to be less than 2%. The relative density of the sintered pellets of SZx and SZYx was 95 ± 2% and 93 ± 2%, respectively.

The electrical resistance of the samples was measured using two-probe AC impedance spectroscopy (Parstat 2273-SVS, Advanced Measurement Technology Inc., Oak Ridge, TN, USA) in a frequency range of 0.1 Hz–1 MHz with an amplitude of 30 mV. For the impedance measurements, platinum paste was applied to the opposite surfaces of the pellets and heated to 1000 °C in air. The electrical resistance was studied in the temperature range of 300–800 °C in air with different humidity (pH_2_O = 40, 610 and 4250 Pa). The value of pH_2_O was held constant by passing the gas flow through a column filled with zeolites (40 Pa) or through a water bubbler kept at 0 °C and 30 °C (610 and 4250 Pa). Prior to the measurements, the samples were equilibrated with the ambient gas for 24 h. First, the measurements were performed at pH_2_O = 40 Pa, then at 610 Pa, and finally at 4250 Pa. In each atmosphere, the impedance spectra were measured under cooling from 800 to 300 °C. The software EQUIVCRT 4.51 was used for the fittings of the measured Nyquist plots [16,17]. Distribution of relaxation times (DRT) was used to identify the bulk and grain boundary contributions to the total conductivity. For DRT analysis, a program code developed in [18], based on Tikhonov’s regularization [19] was applied.

## 3. Results and Discussion

### 3.1. Microstructure, Phase and Chemical Composition of SZx and SZYx

XRD patterns of SZx powders are shown in Figure 1a. For the compositions with low Sr deficiency (x ≥ 0.98), the characteristic peaks can be indexed on the basis of SrZrO_3_ orthorhombic structure as confirmed by the corresponding ICDD file (44-0161). For the compositions with x < 0.98, a weak peak at 2θ = 27.8° which can be attributed to ZrO_2_ monoclinic structure (ICDD 74-0815) appears in addition to the SrZrO_3_ pattern (see the inset in Figure 1a).

Figure 1b presents XRD diffractograms of SZYx powders. As well as in the previous case, most of the peaks can be attributed to an orthorhombic structure of SrZrO_3_, but for the strongly Sr-deficient samples (x < 0.98), a weak peak at 2θ = 29.3° appears to be caused by precipitation of yttrium zirconium oxide (ICDD 89-5593) (see the inset in Figure 1b).

The unit cell volume as a function of Sr content for undoped and Y-doped strontium zirconate is presented in Figure 2. As can be seen, the volume of SZYx unit cell is bigger than that of SZx that can be explained by the lattice expansion under thr introduction of Y^3+^ in Zr^4+^ sites (the ionic radii of Zr^4+^ and Y^3+^ ions with the coordination number six are 0.72 and 0.9 Å, respectively). Besides, the unit cell volume of SZx and SZYx increases with decreasing Sr content up to x = 0.96 and 0.98, respectively, and then stays nearly constant. This observation indicates that a solid solution of strontium vacancies in strontium zirconate crystal structure forms at low Sr deficiency (x > 0.96 and 0.98 for SZx and SZYx, respectively), whereas precipitates of a second phase start to form at lower x values.

Figure 3a–d shows SEM images of SZx surface after polishing and etching at 1400 °C. For the Sr-deficient compositions, small grains of the second phase localized at the grain boundaries of the main phase can be observed. A chain of precipitated nanograins can also be seen in the fracture SEM image of SZ0.96 specimen presented in Figure 3e. EDX analysis performed on the surface of SZx has shown that chemical composition of the surface of the main phase grains is nearly the same for all values of x being close to Sr_0.95±0.01_ZrO_3−δ_, and that of the second phase grains is close to ZrO_2_. The latter correlates with the XRD data showing the reflections of ZrO_2_ (Figure 1a).

The fact that the surface layer of SZx ceramics is depleted with strontium comparing to the nominal composition can be caused by evaporation of strontium. Since SrO and BaO exhibit high vapor pressure, their evaporation from cerates and zirconates of the alkaline earth elements can occur. To prevent the evaporation, special efforts, such as sintering in a sacrificial powder bed or using of Ba-saturated furnace furniture, should be made [20,21,22]. Therefore, we assume that strontium evaporation occurs in strontium zirconate during the high-temperature sintering and thermal etching. Therefore, the observed precipitation of ZrO_2_ grains on the surfaces of SZ0.98 and SZ1.00 whose compositions belong to the range of solid solution formation (see Figure 3c,d) can be caused by enhanced Sr deficiency in the surface layer of the ceramics.

Comparative analysis of the SEM data for SZx samples reveals that nominal deficiency of strontium stimulates the growth of the main phase grains from 1–2 μm in the sample with a stoichiometric strontium content to about 5 μm for the Sr-deficient samples. Therefore, deficiency of strontium enhances grain growth in strontium zirconate.

SEM study of the surface of SZYx after polishing and etching at elevated temperatures also revealed the second phase precipitation in the Sr-deficient samples. For illustration, the SEM images and EDX mapping of the elements on the surface of SZY0.94 after polishing and etching at 1400 and 1650 °C are presented in Appendix A. The grains of a light grey color with ~500 nm diameter embedded in the large grain matrix of a dark grey color are observed in the SEM pictures. As it follows from the EDX mapping, strontium, zirconium and yttrium are homogeneously distributed over the large grain matrix. The light grey grains contain mostly zirconium and yttrium in the ratio close to 1:1. This observation is consistent with the XRD data showing the reflexes of yttrium zirconium oxide for SZYx with x = 0.94 and 0.96 (Figure 1b).

The second phase is assumed to be Y_0.5_Zr_0.5_O_2_ having a defect fluorite structure as far as pyrochlore zirconates A_2_B_2_O_7_ were shown to exist in the range of A^3+^/B^4+^ cation radius ratio of 1.46–1.78 [23], and for Y_2_Zr_2_O_7_ this ratio equaled to 1.42 does not belong to the stability field of pyrochlores. The size of the main phase grains in SZYx varies from 1–2 μm for x ≥ 1.00 to 8–10 μm for the Sr-deficient compositions.

As can be seen in Appendix A, increasing the etching temperature from 1400 to 1650 °C results in growth of the second phase grains (from ~500 nm to ~2 μm) that can be explained by diffusion of yttrium and zirconium along the grain boundaries from the interior of the sample or/and by enhanced strontium depletion of the surface layer because of more intensive evaporation.

The surface composition of strontium zirconate grains was found to be close to Sr_0.96±0.01_ZrO_3−δ_ for all SZYx compositions. An unexpected result shown by the EDX data is that the surface layer of strontium zirconate grains does not contain detectable amounts of yttrium. In order to obtain information about the chemical composition of the interior of SZYx ceramics the as-polished (non-etched) cross-sections of several samples—SZY0.98, SZY1.00 and SZY1.02—were examined by SEM-EDX. After polishing the surface layer of grains is removed, so the non-etched polished surface should possess the composition close to that of the grain interior. Thermal etching activates recrystallization of grains leading to a decrease in surface energy, and the grain boundaries become more visible. Besides, thermal etching activates diffusion of the elements that leads to change in the surface composition. The etching at the temperature of sintering should result in the restoration of the surface composition of the sintered samples. As can be seen in Appendix A the grain topography of the polycrystalline samples is less distinguishable without thermal etching. The average concentration of yttrium over the cross-section was found to be less than the nominal one, being close to 2 mol%. Low yttrium concentration in the interior of strontium zirconate grains is consistent with the observed precipitation of the Y-containing phase. 

At first glance, the low solubility of yttrium in strontium zirconate revealed in the present research does not match the previously reported data [11,13,24,25]. Huang and Petric [11], for example, reported on the formation of single-phase solid solutions in SrZr_1−x_Y_x_O_3−δ_ synthesized via solid-state reaction and sintered at 1500–1600 °C between x = 0 and 0.2, however, the XRD data were not presented.

Films and pellets of SrZr_0.84_Y_0.16_O_3−δ_ were studied in [24]. The films were deposited by co-sputtering of Sr and Zr_0.84_Y_0.16_ metallic targets in the presence of argon and oxygen gas mixture. The pellets were synthesized via a self-combustion technique employing glycine as fuel and nitrates of metal components as oxidants. The disc-shaped samples were uniaxially pressed at 125–150 MPa and sintered at 1700 °C for 5 h in air. The ratio of (Zr + Y)/Sr determined by EDX was reported to scatter from 0.91 for the films to 1.07 for the pellets, which indicates that there is a considerable deviation between the real and nominal compositions. Besides, the X-ray diffractogram of the pellet presented in [24] contains unidentified minor lines indicating the formation of additional phases in SrZr_0.84_Y_0.16_O_3−δ_.

Thus, the phase composition of SrZrO_3_ heavily doped with yttrium appears to be more complicated than expected. Strontium zirconate with a smaller concentration of yttrium was studied in [13,25]. Single crystals of SrZrO_3_ nominally doped with 5 mol% yttrium were grown by a floating zone method in an oxygen-containing atmosphere [13]. Chemical analysis using atomic emission spectroscopy yielded the composition (Sr_0.99_Y_0.01_)(Zr_0.96_Y_0.04_)O_2.97_ assuming that all cation sites are occupied and all cations are in their highest oxidation state. Possibly, the high temperature needed for the single crystal growth ensures the higher solubility of yttrium than it was reached in our research.

Polycrystalline SrZrO_3_ doped with 5 mol% yttrium was obtained by Pechini method, the isostatically pressed pellets were sintered at 1700 °C for 20 h in air [25]. XRD pattern obtained in Cu Kα radiation showed a set of peaks characteristic for SrZrO_3_ with an orthorhombic structure along with a minor peak at 2θ ~ 29.3° which was ignored. Therefore, under careful consideration, the data on yttrium solubility in strontium zirconate obtained in the present research do not contradict to those reported for the polycrystalline samples.

Summarizing the results of XRD, SEM and EDX study of SZx and SZYx ceramics one can conclude that:(i)at low Sr deficiency, solid solutions of strontium vacancies in the crystal structure of SZx and SZYx form (at x > 0.96 and 0.98, respectively), whereas at higher Sr deficiency the secondary phase, zirconium or yttrium zirconium oxide, is precipitated;(ii)yttrium solubility limit in SrZrO_3_ sintered at 1650 °C is close to 2 mol%, the excess yttrium is included in the secondary phase;(iii)Sr-deficiency enhances grain growth in SZx and SZYx ceramics.

### 3.2. Electrical Conductivity of SZx and SZYx

The electrical conductivity of SZx and SZYx was determined by deconvolution of the impedance spectra. For illustration, the spectra for SZY0.98 and SZY1.00 at 550 °C are given in Figure 4. The Nyquist plots for SZY1.00 exhibit three semicircles: the first semicircle is due to the contribution of bulk properties of the material at high frequency, the second is due to the grain boundaries at a medium frequency, and the third is due to electrode polarization. Nyquist plots were modeled by the equivalent circuit (R_b_Q_b_)(R_gb_Q_gb_)(R_el_Q_el_) composed of three parallel RQ elements (R is a resistor, Q is a constant phase element) connected in series. The resistors R_b_, R_gb_ and R_el_ represent bulk, grain boundary and electrode polarization resistances, respectively. 

For SZY0.98, the spectrum consists of two semicircles. The low-frequency process with a capacitance of 10^−5^ F/cm^2^ can be ascribed to the electrode polarization. The high-frequency arc is significantly depressed which is typical for the grain boundary response. At low temperatures (300–450 °C) this arc possesses a high-frequency shoulder related with the growing response of the grain bulk. The Nyquist plots for SZY0.98 were modeled by the equivalent circuit R_b_(R_gb_Q_gb_)(R_el_Q_el_) in the temperature range of 500–800 °C. At the temperatures below 450 °C, DRT technique was applied to identify the bulk and grain boundary contributions. This technique helps to resolve the components of a system with close time constants or a significant difference between resistances.

For comparison, the high-frequency DRT data at 350 °C (pH_2_O = 40 Pa) for SZY0.98 and SZY1.00 are presented in Figure 5. It can be seen that DRT curves for both compositions demonstrate maximums at about 600 Hz and 10 kHz, which can be related to the responses of grain boundaries and grain bulk, respectively. DRT curves show that the grain boundary responses of SZY0.98 and SZY1.00 are comparable, whereas the bulk peak for SZY0.98 is much less than that for SZY1.00, that means that the bulk resistance of SZY0.98 is smaller. The differences in the behavior of SZY0.98 and SZY1.00 can be caused by changes in the grain size and Sr content. The microstructural features affect mostly the grain boundary resistance, whereas the chemical composition should influence both bulk and grain boundary resistance. The behavior of the bulk and grain boundary resistance of SZYx will be discussed in more detail below.

The total conductivities of SZx and SZYx obtained as described above in Arrhenius coordinates are linear as can be seen in Figure 6. The activation energies of the conductivity for each composition are also given in the figure. The total and bulk conductivities of SZx and SZYx as functions of strontium content in air (pH_2_O = 40 Pa) are presented in Figure 7. As can be seen, the conductivity of SZx is less than in SZYx, and the conductivity of both doped and undoped strontium zirconate is sensitive to strontium content. The dependences of the total and bulk conductivity on Sr content possess a maximum at x = 0.98 for both systems.

It is known that undoped alkaline earth metal zirconates possess low ionic conductivity because of a small degree of their own disordering [9,10,26]. Acceptor-doping results in the appearance of oxygen vacancies that leads to increasing ionic transport:
(1)$$Y_{2}O_{3}\to 2Y_{Zr}^{/}+V_{O}^{••},$$
where $Y_{Zr}^{/}$ is a substitution defect and $V_{O}^{••}$ denotes an oxygen vacancy.

Additional oxygen vacancies should appear due to Sr-deficiency in SrZrO_3_:
(2)$$-SrO\to V_{Sr}^{//}+V_{O}^{••}$$
where $V_{Sr}^{//}$ is a vacancy of strontium.

For electrical neutrality, the concentrations of the cation and anion vacancies should obey the following equation:(3)$$\left[V_{Sr}^{//}]+2[Y_{Zr}^{/}]=2[V_{O}^{••}\right]$$

The defect model expressed by Equations (1)–(3) shows that both acceptor doping in B sites and A-site cation deficiency should result in the increasing oxygen vacancy concentration that gives rise to the ionic conductivity. The model describes well the experimental data on the conductivity of SZx and SZYx at x > 0.98: Firstly, as can be seen in Figure 6 and Figure 7 yttrium doping results in increased conductivity of SZYx comparing to SZx due to higher concentration of oxygen vacancies. Secondly, Sr deficiency of up to x ~ 0.98 creates additional oxygen vacancies as it follows from XRD data (Figure 2), so that SZ0.98 and SZY0.98 have the highest conductivity. Overstoichiometric nominal strontium content (x = 1.02) leads to decrease in conductivity which might result from the decreasing concentration of oxygen vacancies. 

A possible reason for the observed decrease in the conductivity of SZx and SZYx with high Sr deficiency (x < 0.98) is the low conductive phases precipitation. Another reason can be a dopant partitioning over A and B-sites in ABO_3_ perovskite lattice. The incorporation of trivalent dopant onto the Sr sites should have a detrimental effect on the conductivity because of a decrease of the oxygen vacancy concentration which can be described as follows:
(4)$$\left[V_{Sr}^{//}]+2[Y_{Zr}^{/}]=2[Y_{Sr}^{•}]+[V_{O}^{••}\right]$$
where $Y_{Sr}^{•}$ is yttrium ion in Sr site.

The effect of the dopant partitioning was reported in a number of researches [15,22,27,28,29]. The ionic size of the dopant was shown to have an impact on its localization in perovskite lattice. Rietveld refinement of X-ray diffraction data of BaZr_0.8_M_0.2_O_3−δ_ (M = Sc, Eu, Sm, Dy) revealed that Eu, Sm, and Dy cations occupied both A-and B sites of BaZrO_3_ crystalline lattice, whereas Sc cations were determined to occupy B sites only [27]. These results indicate clearly an increasing tendency toward A-site occupation for the dopant cations in BaZrO_3_ with an increasing radius. The dopant partitioning over both A-and B sites in yttrium-doped barium zirconate was assumed to be responsible for a decrease of the bulk conductivity [15].

Infrared absorption spectroscopy study of SrZr_0.95_M_0.05_O_3−δ_ (M = Ga, Sc, In, Lu, Y and Gd) revealed two types of proton site, M-OH-Zr and M-OH-M, for all compositions except for Sc whose ionic radius is the closest to that of the host Zr [28]. Presence of two types of proton sites at low dopant concentration (5 mol%) indicates that the dopant ions are not statistically distributed over the B-site positions in SrZr_0.95_M_0.05_O_3−δ_ but partially accumulated in MO_6_-clusters. It was supposed that the structural distortion introduced by doping due to the size difference of the dopant M compared to Zr is the driving force for MO_6_-cluster formation [28].

First-principles calculations performed for Y-doped barium zirconate [22] also showed that there were Y ions residing on the Zr and Ba sites. Besides, it was concluded that local enrichment of Y in bulk and on the surface observed by X-ray photoelectron spectroscopy attained a structure similar to Y_2_O_3_ that can be considered as precipitation of the yttrium-containing phase. 

Weston et al. [29] studied Y and Sc dopants in SrZrO_3_ using first-principles density functional theory calculations with a hybrid functional. Sc and Y were shown to incorporate on Zr sites, forming deep acceptor states and triggering the formation of oxygen vacancies. However, they can also incorporate on the Sr sites, where they form donor states. It was found that incorporation on Sr sites is a more serious problem for larger Y ions than for scandium.

So, the probability of Y incorporation onto Sr sites in SZYx cannot be ignored and increasing Sr deficiency is expected to promote the formation of defects. In this case, the electrical conductivity should decrease with increasing Sr deficiency as it is observed at x < 0.98 (Figure 7). 

It follows from Equation (3) that in undoped SrZrO_3_ concentration of oxygen vacancies is determined by A-site vacancies:
(5)$$\left[V_{Sr}^{//}]=[V_{O}^{••}\right]$$

If in the Y-doped SrZrO_3_ Y ions predominantly incorporate onto Sr sites, i.e., $\left[Y_{Zr}^{/}]<[Y_{Sr}^{•}\right]$, then it follows from Equations (4) and (5) that the oxygen vacancy concentration in SZYx will be less than in SZx. So, Y ions partitioning over both A and B sites explains why the conductivity of strongly Sr-deficient Y-doped strontium zirconate, SZY0.94, becomes lower than that of SZ0.94.

The dopant partitioning also explains the behavior of the unit cell volume under yttrium doping shown in Figure 2. As it was discussed, at low Sr deficiency, the volume of SZYx unit cell is bigger than that of SZx because of the lattice expansion under substitution of small Zr^4+^ ions by large Y^3+^ ions. However, at high Sr deficiency, the lattice expansion is compensated by the contraction due to yttrium incorporation onto the Sr sites, as a result, the unit cell volumes of SZYx and SZx are getting close.

The conductivity of grain boundaries of SZYx also depends on Sr stoichiometry exhibiting the highest values at x = 0.98 and 1.00 as can be seen in Figure 8. The grain boundary conductivity behavior does not correlate with the grain size change in SZYx which indicates that other factors have a greater impact on it. Increasing of the conductivity with decreasing Sr content at high x (x > 0.98) can be caused by formation of additional oxygen vacancies for charge compensation of A site vacancies. Decreasing of the conductivity at lower x can be caused by the blocking effect of the low conductive phase precipitated at the grain boundaries, besides, the dopant partitioning over A and B sites can also occur at the grain boundaries.

As can be seen in Figure 9 which demonstrates Arrhenius plots of the bulk and grain boundary conductivity of SZYx in air with different humidity, the conductivities increase with pH_2_O. For comparison, the conductivity of a single-crystalline sample with a nominal composition of SrZr_0.95_Y_0.05_O_3−δ_ reported in [30] is also shown. A good correlation between the conductivity values can be seen.

For the generation of the proton charge carriers, the presence of oxygen vacancies in the crystal lattice is required. The oxygen vacancies can be created by acceptor doping and/or deficiency of A-site cations in strontium zirconate (Equation (3)). Dissolution of water vapor in the crystal lattice can be written as:
(6)$$H_{2}O(g)+V_{O}^{••}+O_{O}^{\times }=2OH_{O}^{•}$$
where $OH_{O}^{•}$ denotes a proton localized on the oxygen ion.

It follows from the Equation (6) that the proton defects concentration can be written as:(7)$$[OH_{O}^{•}]={\left(K_{6}[V_{O}^{••}]\right)}^{1/2}{\left(pH_{2}O\right)}^{1/2}$$
or, taking into account the electroneutrality condition (4):(8)$$[OH_{O}^{•}]={\left(K_{6}[V_{Sr}^{//}]\right)}^{1/2}\frac{\left[Y_{Zr}^{/}\right]}{\left[Y_{Sr}^{•}\right]}{\left(pH_{2}O\right)}^{1/2}$$
where *K*_6_ is an equilibrium constant of the Equation (6).

Taking into account Equation (8) and assuming that the concentrations of defects, [$V_{Sr}^{//}$], $\left[Y_{Zr}^{/}\right]$ and $\left[Y_{Sr}^{•}\right]$, do not depend on pH_2_O, the proton conductivity which is proportional to the concentration of proton defects [$OH_{O}^{•}$] has to increase with the partial pressure of water vapor:
™_H_ ~ (pH_2_O)^1/2^(9)

Dependences of the bulk and grain boundary conductivity of the most conductive samples, SZY0.98 and SZY1.00, on water vapor partial pressure in logarithmic scale at 350 °C are shown in Figure 10. As can be seen, the slope of the dependences is close to ½ being in agreement with the suggested model of proton defect formation (Equation (9)).

The activation energies of the bulk and grain boundary conductivity are found to depend on Sr content having the lowest value at x = 0.98 and decrease with a rise in water vapor partial pressure as shown in Figure 11. The values of activation energy obtained in the present research are consistent with the reported data for SrZr_0.95_Y_0.05_O_3−δ_ in wet nitrogen [25] which are also given in Figure 11. The higher values of the activation energy of the grain boundary conductivity comparing to the bulk indicate the existence of an additional energy barrier which can be related with the formation of space charge layers at the grain boundaries [31].

## 4. Conclusions

Effect of Sr nonstoichiometry on phase composition, microstructure, defect chemistry and electrical conductivity of Sr_x_ZrO_3−δ_ and Sr_x_Zr_0.95_Y_0.05_O_3−δ_ ceramics (SZx and SZYx, respectively; x = 0.94–1.02) was investigated via X-ray diffraction, scanning electron microscopy, energy-dispersive X-ray spectroscopy and impedance spectroscopy followed by distribution of relaxation times analysis of spectra. SZx and SZYx were successfully synthesized by a chemical solution method. It was shown that at low Sr deficiency, solid solutions of strontium vacancies in SZx and SZYx form (at x > 0.96 and 0.98, respectively), whereas at higher Sr deficiency the secondary phase, zirconium oxide or yttrium zirconium oxide, is precipitated. Yttrium solubility limit in SrZrO_3_ was found to be close to 2 mol%. It was shown that strontium deficiency enhances grain growth in the strontium zirconates studied.

A-site nonstoichiometry and yttrium doping were found to have a significant effect on the electrical conductivity of strontium zirconate. The highest total and bulk conductivities were observed at x = 0.98 for both systems. Y-doped strontium zirconate possesses up to two orders of magnitude higher total conductivity than Sr_x_ZrO_3−δ_ samples. In Y-doped strontium zirconate, dopant partitioning over A and B sites is assumed to be the main reason for a decrease of conductivity at high Sr-deficiency. The conductivity of SZYx was found to increase with a rise in humidity, which indicates the appearance of proton conduction in wet conditions. A defect model based on consideration of different types of point defects—strontium vacancies, substitutional defects and oxygen vacancies, and assumption of Y ions partitioning over Zr and Sr sites, was elaborated; the proposed model consistently describes the obtained data on conductivity.

## Figures and Tables

**Figure 1 materials-12-01258-f001:**
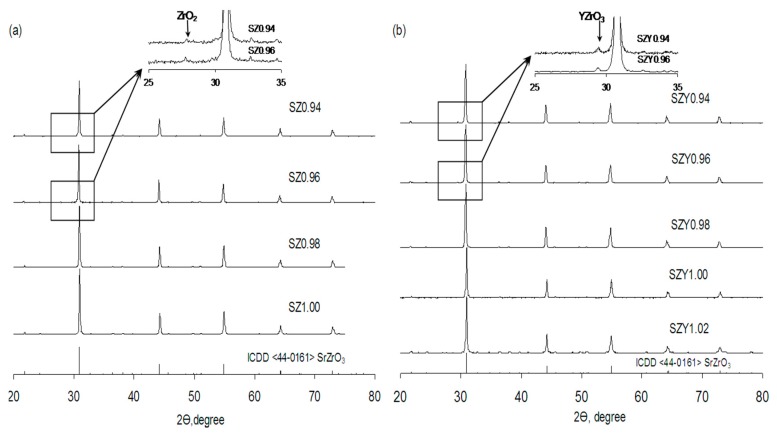
Powder XRD patterns of (**a**) SZx and (**b**) SZYx. The insets show fragments of diffractograms in the 2θ angles from 25° to 35° for x = 0.94 and 0.96.

**Figure 2 materials-12-01258-f002:**
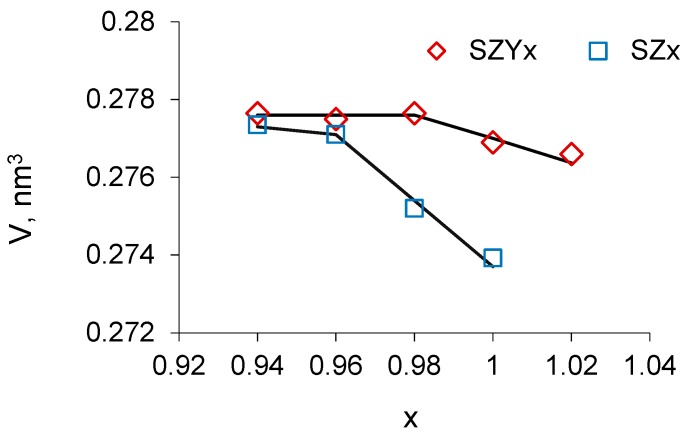
Unit cell volume as a function of Sr content in SZx and SZYx.

**Figure 3 materials-12-01258-f003:**
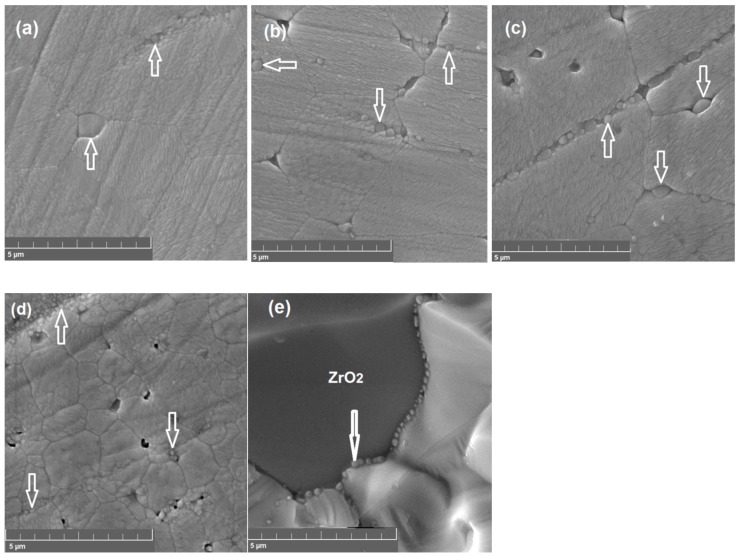
SEM images of the surface of SZx after polishing and etching at 1400 °C: (**a**) SZ0.94, (**b**) SZ0.96, (**c**) SZ0.98, (**d**) SZ1.00, and (**e**) fracture surface of SZ0.96. ZrO_2_ grains are marked by arrows.

**Figure 4 materials-12-01258-f004:**
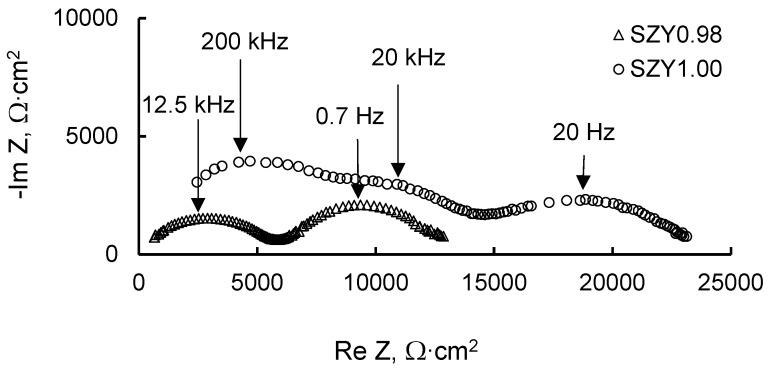
Impedance spectra of SZY0.98 and SZY1.00 samples at 550 °C in air (pH_2_O = 40 Pa).

**Figure 5 materials-12-01258-f005:**
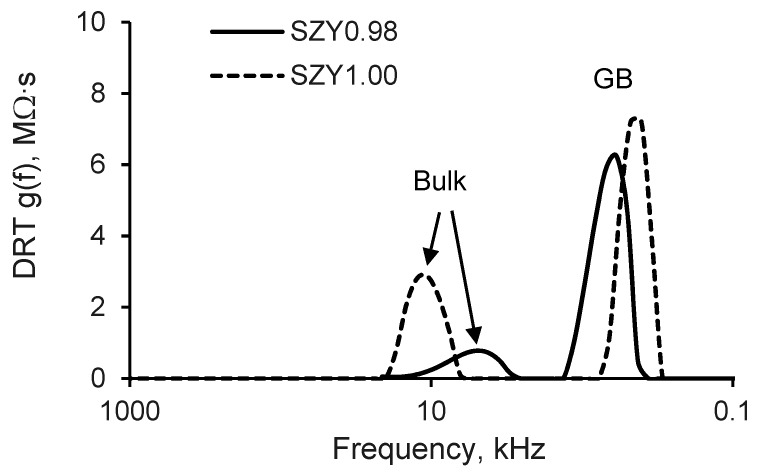
DRT data for SZY0.98 and SZY1.00 at 350 °C in air (pH_2_O = 40 Pa).

**Figure 6 materials-12-01258-f006:**
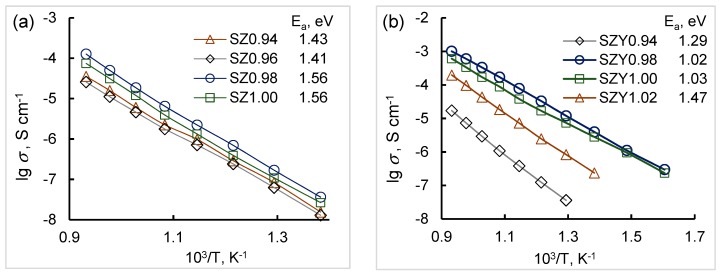
Arrhenius plots of the total conductivities of SZx (**a**) and SZYx (**b**) in air (pH_2_O = 40 Pa). The lines between points are only a guide to the eye.

**Figure 7 materials-12-01258-f007:**
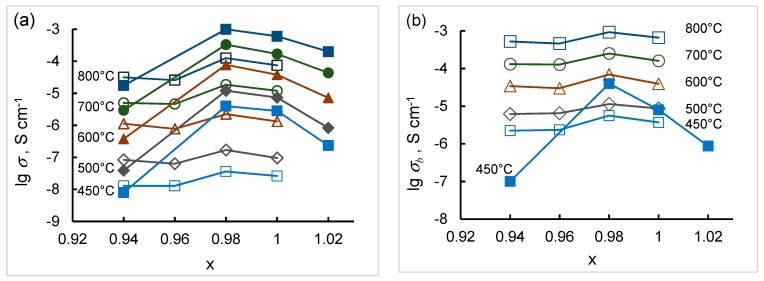
Total (**a**) and bulk (**b**) conductivity of SZx (empty markers) and SZYx (solid markers) as a function of strontium content in air (pH_2_O = 40 Pa). The lines between points are only a guide to the eye.

**Figure 8 materials-12-01258-f008:**
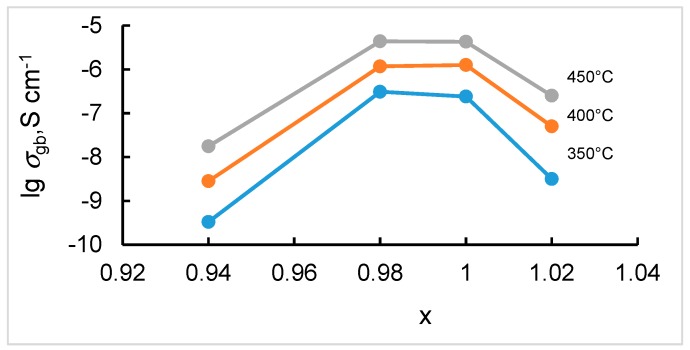
Grain boundary conductivity of SZYx as a function of strontium content in air (pH_2_O = 40 Pa). The lines between points are only a guide to the eye.

**Figure 9 materials-12-01258-f009:**
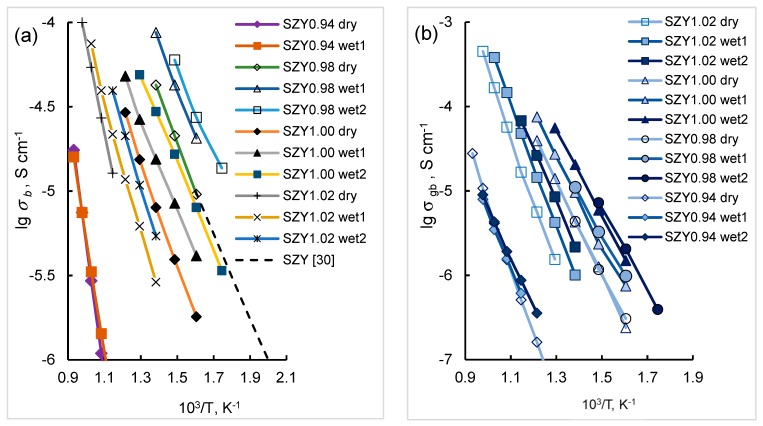
Arrhenius plots of bulk (**a**) and grain boundary (**b**) conductivity of SZYx in dry (pH_2_O = 40 Pa) and wet air (wet1—pH_2_O = 610 Pa, wet2—pH_2_O = 4250 Pa). The lines between points are only a guide to the eye. Conductivity of a single-crystalline SrZr_0.95_Y_0.05_O_3−δ_ measured in dry air [30] is given for comparison.

**Figure 10 materials-12-01258-f010:**
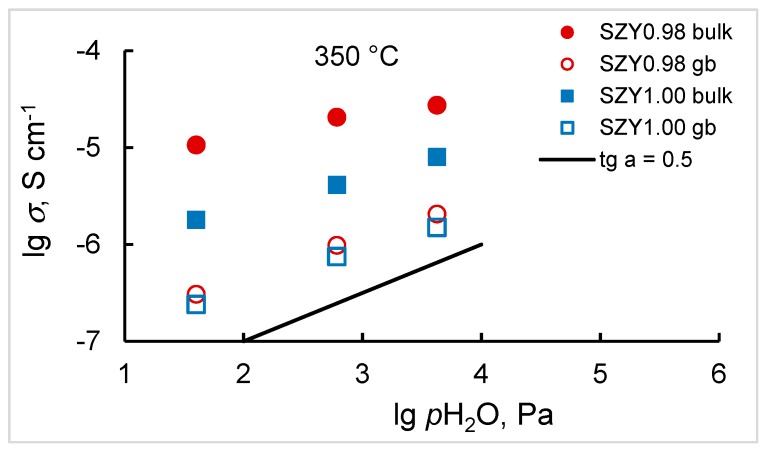
Bulk and grain boundary conductivity vs. humidity for SZY0.98 and SZY1.00 at 350 °C.

**Figure 11 materials-12-01258-f011:**
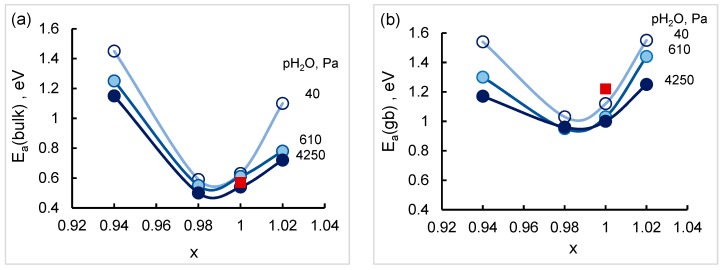
Activation energies of the bulk (**a**) and grain boundary (**b**) conductivity of SZYx. The lines between points are only a guide to the eye. Activation energies for SrZr_0.95_Y_0.05_O_3−δ_ measured in wet nitrogen [25] are given for comparison (red squares).

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
