# Peer review of "Effect of A-Site Nonstoichiometry on Defect Chemistry and Electrical Conductivity of Undoped and Y-Doped SrZrO3"

_materials, 2019, doi:10.3390/ma12081258_

Round 1
Reviewer 1 Report
In general the data presented in this paper is well collected and is interesting for the SOFC community regarding
the need for highly conductive electrolyte materials. However the quality of the presentation of the data in the paper
text is poor and authors should rearrange the data presented and move some part to supporting information section.
Also the extensive work on the English is required.
Authors should carefully revise the text and provide more matured version.
1. My main concern is about unreasonable paper length of 19 pages and 14 figures even the actual amount
of scientific data presented is not so significant i.e. just SEM, XRD, DRT and EIS results. At the same time
such type of data presentation is negatively affects readability of the text. Authors should carefully revise work
and re-consider the paper structure and move some parts so supplementary material section. General papers
with similar content are recommended to have of 10 or less figs. and be of less than 10 pages.
2. As an example I propose to unite Figs 1, 2, 3 and move 1b and 2b to SI.
3. Same for SEM. Fig 4 occupies the whole page. How about show higher magnification images but combine
the whole a-e figures within one line?
4. The message of Fig 6 is unclear. Why “the grain structure of the polycrystalline samples is nearly
indistinguishable without thermal etching”? At the image grains of SZY are clearly seen, at the same time it is
also possible to see small particles of precipitated phases. Thus this image provides very similar information as
figs 4 and 5.
5. Conductivity plots. Figs 9, 10, 12, 14. Why label a and b are in the top-middle area of figures. Typically those
labels are places in the top-left part.
6. In abstract and conclusion authors mentioned achieved 2 orders in increase in conductivity but from the data
presented it is not clear which conductivity authors mentioned, bilk, grain, or in the presence of vapor. Should be
clarified.
7. Finally English should be improved across the text. Here are only several example of non-senses in the text,
but there are much more.
Line 15. “followed by distribution of relaxation times analysis of spectra”. Sentence is unclear. Which spectra? EDS or EIS?
Should be clarified.
Line 17 “…strontium vacancies form a solid solution in SZx and SZYx”. How can oxygen vacancies form solid solution?
Nonsense, please check definition of “solid solution”. Same at line225.
Line 23. “ Increasing the conductivity with a rise in humidity indicates to proton transport in the oxides”. Sentence should
be revised. Increase in conductivity in wet conditions cannot indicate to solely proton transport phenomena. Sentence
reads like oxygen transport phenomena is absent.
Next sentence. “A defect model consistently describing the obtained data on conductivity was elaborated.” Does authors
mean “A defect model of conductivity mechanisms based on obtained data was elaborated”? I am not sure what authors
mean but at least such configuration the sentence has sense.
Lines 144-155. “Evaporation of SrO and BaO having high vapor pressure…”. How “evaporation” can have “vapor pressure”?
Sentence should be revised.
Line 191. “…etching leads to yttrium depletion of the surface layer of strontium zirconate grains”. Nonsense.
Again those are only several examples and there are much more in the text.
Author Response
Thank you for your helpful comments. We have revised the manuscript accordingly and feel that your comments helped
clarify and improve our paper.

Reviewer 2 Report
1) Instead of writing x> 0.96-0.98, please write either 0.96<x<0.98 or x=0.96 and 0.98.
2) Why did the authors choose thermal etching at 1650°C, which is the sintering temperature? It may have increased
the grain size. I cannot follow the reasoning for this, please discuss in the text.
3) How did the authors get the unit cell parameters? The fitting program (FullProf etc.) and the goodness of fit need to
be written. As the density of pellets is estimated from the theoretical density, the estimation of the porosity would have an
influence on the electrical conductivity values. Secondly, does the density calculation give similar values by Archimedes
method?
4) From figure 1, the intensity ratios between the 2nd and the 3rd highest intensity peaks change when x<0.98. As I don’t
see clearly a change in the peak positions, it may indicate a textured growth. This is also supported by the fact that the XRD
is taken on ground sintered pellets. However, texture growth is also accompanied by the change in FWHM (more narrow or
broader peaks, which is also difficult to observe in the figure).
It may also indicate a change in symmetry related to defect ordering. For that reason, it is very important to study the lattice
structure and lattice parameters very well. Later in the text, the authors discuss the possibility of dopant partitioning. Can it
explain the intensity change? I am not sure how reliable it would be to compare the effect of dopant partitioning by volume
change as it is not described how the unit cell parameters are obtained. I believe it is indispensable to study the XRD data
more appropriately by Rietveld refinement.
5) The paragraph starting at line 181 discusses the surface composition of SZYx and the amount of yttrium at the interiors
of the unetched pellet. How about the interiors of the etched sample or the surface of the unetched sample? Is it less than
2% as well? If there is a possibility of doing elemental analysis on the initial powder as well as scratched particles from the
etched and unetched particles, it would help to explain the low concentration of yttrium.
6) The authors need to comment on the DRT intensity ratios for bulk and grain boundary contributions. What does it signify
having much smaller intensity for bulk SZY0.98 than SZY1.0 Are they related to grain size?
7) The activation energy values in Fig. 9 and 12 should be given in the graph.
8) Are all the conductivity measurements done during heating or cooling cycles? Is there any difference?
9) Please make sure that you have the same abbreviation for logarithm scale. In some graphs, it is log, in some of them,
you have written lg.
10) As I understand, the bulk conductivity values of SZYx above 450C was not possible to obtain (Fig 10). Then give the total
conductivity values at 450C in Fig.10a as well.
11) Eq 3 in line 283 misses a x2 before the oxygen vacancy concentration. Eq.4 in line 296 does not follow Kroger-Vink notation.
Strontium vacancy is charged -2, yttrium ion in zirconium site is -1 and yttrium in strontium site is +1. I cannot follow how this
equation is electroneutral.
12) In Fig 12a, is the y scale multiplied by temperature? Please make sure all the graphs are coherent.
13) Fig 14, please give the same scales at the axes in a) and b), it is visually easier for the readers.
14) Nowhere in the text, the effect of excess Strontium is discussed. Some comments are necessary.
15) The language needs to be polished. There are certain words that need replacing in the following:
-Line 46 perhaps (e.g. attributed to)
-Line 50 supposedly
-Line137…precipitated along the grain boundary
-Line 172 is supposed to need to be replaced, such as ‘may adopt’. Similarly on line 304, supposed to be can be replaced with
proposed to be, assumed to be etc.
Line 209-210: ..”seems to be more complicated than it was supposed to be” could be replaced by “appears to be more complicated
than expected”.
-Line243 and 245. Please change the two strong words ‘definitely’ and ‘obviously’.
-Line 286 ‘either’ doesn’t make sense in this sentence. Do you mean ‘should result in an increase of ionic conductivity followed by
an increase in the oxygen vacancy concentration.’?
-Line 314: both comes before yttrium.
You may refer the precipitates as secondary or minority phase.
All over the text: indicates to, it should be only indicates (delete to).

Author Response

(The authors gave the same response as above.)

Round 2
Reviewer 1 Report
The paper is suitable for publication now